# Selective Oxidation of Isobutane to Methacrylic Acid and Methacrolein: A Critical Review

**Li Zhang** , **Sébastien Paul** , **Franck Dumeignil \*** **and Benjamin Katryniok \***

University Lille, CNRS, Centrale Lille, University Artois, UMR 8181–UCCS–Unité de Catalyse et Chimie du Solide, F-59000 Lille, France; li.zhang@centralelille.fr (L.Z.); sebastien.paul@centralelille.fr (S.P.)
\* Correspondence: franck.dumeignil@univ-lille.fr (F.D.); benjamin.katryniok@centralelille.fr (B.K.)

**Abstract:** Selective oxidation of *iso*butane to methacrolein (MAC) and methacrylic acid (MAA) has received great interest both in the chemical industry and in academic research. The advantages of this reaction originate not only from the low cost of the starting material and reduced process complexity, but also from limiting the use of toxic reactants and the production of wastes. Successive studies and reports have shown that heteropolycompounds (HPCs) with Keggin structure (under the form of partially neutralized acids with increased stability) can selectively convert *iso*butane to MAA and MAC due to their strong and tunable acidity and redox properties. This review hence aims to discuss the Keggin-type HPCs that have been used in recent years to catalyze the oxidation of *iso*butane to MAA and MAC, and to review alternative metal oxides with proper redox properties for the same reaction. In addition, the influence of the main reaction conditions will be discussed.

**Keywords:** Keggin-type HPCs; *iso*butane; oxidation; methacrolein; methacrylic acid





## 1. Introduction

Hydrocarbons are generally issued from coal, oil, and natural gas. They are still the main feedstock of the chemical industry. In addition, natural gas extraction has become increasingly significant as, compared to other fossil energies, its supply is expected to last longer than that of oil (roughly 200 years in the Middle East taking into account the currently known reserves and consumption) [1]. The transformation of hydrocarbons constitutes an extremely important field of industry, and consequently of research, whereby notably selective oxidation is among the most important chemical processes. Many chemical products and fine chemistry intermediates are actually synthesized via a process that includes an oxidation step [2–5]. However, only a portion of the chemical processes for the oxidative conversion of light alkanes have been industrialized, despite the fact that they are cheap and widely available in natural gas and liquefied petroleum gas. The oxidative conversion of light alkanes is a much more complex technological challenge than the conversion of corresponding olefins due to their poor chemical reactivity. Methane conversion was reported to be only 1.5% and formaldehyde selectivity 40% over $VO_x$/SBA-15 catalyst (at a reaction temperature of 618 °C); ethane conversion was also only 1.5% and ethylene selectivity 86% over MoVNb/$TiO_2$ catalyst (reaction temperature of 275 °C) [6,7]. Nevertheless, alkanes are cheaper raw materials than the corresponding olefins, available in abundance and with a low toxicity compared to aromatics [8–12].

Selective oxidation of light alkanes gives access to valuable products. In addition to the reported oxidative dehydrogenation (ODH) of short-chain alkanes to produce the corresponding alkenes [13,14], many oxygenated products, such as acetic acid (AA), acrylic acid (ACA) [15,16], methacrylic acid (MAA) [17], methacrolein (MAC), etc. [18,19], are important chemicals or monomers in the petrochemical industry. Among them, MAA, which has two functional groups: a carbon-carbon double bond and a carboxylic acid group, has attracted much attention. It is used to prepare methyl methacrylate (MMA) for coating, rubber, adhesive, resin, polymer material additives, and functional polymer

materials. MMA, as a specialty monomer for poly-methyl-methacrylate (PMMA), is widely used in civil engineering, medicine, and fine chemicals [20–24]. A dozen technologies are under development or practiced commercially for synthesizing MMA, as the demand for MMA surpassed 4.8 million metric tons in 2020 [21,25]. Constant growth of the MMA market is expected in the near future and therefore an increase in the production of PMMA will be necessary.

Selective oxidation of *iso*butane to MAA, which would provide a one-step route, has attracted extensive attention because of the simplicity of the process and decreased by-product generation [26,27]. Moreover, this route is inexpensive and would have a lower environmental impact than other fossil feedstocks-based routes, as it optimally converts *iso*butane to MAA in a single step [28–31]. Rohm & Haas Company was the first, in 1981, to claim one-step oxidation of *iso*butane to MAC and MAA based on Mo/P/Sb oxides, of which the compositions are close to that of Keggin-type heteropoly acids [32]. Then, several patents appeared in the 1980s and 1990s claiming the possibility of carrying out the synthesis of MAA by the one-step oxidation of *iso*butane in the gas phase, over Keggin-type heteropolyacids as heterogeneous catalysts [33].

However, there are two hurdles to overcome with the oxidation of *iso*butane to MAA and MAC. The first one is to activate the C–H bonds using as low energy as possible to initiate the transformation, and the second one is to selectively generate the targeted compound. Many parameters affect these reactions, and at least two of them are related to the catalyst. Bifunctional catalysts with acidic and redox properties are necessary to activate the C–H bond [34–37], and redox properties also have an effect on the oxygen insertion reaction with lattice oxygen $O^{2-}$, which is known to be more selective for desired products than electrophilic species, such as $O_2^-$ and $O^-$, responsible for degradation, generating overoxidation products. Therefore, bifunctional catalysts simultaneously possessing acid and redox properties are necessary, which brings heteropolyanion-based catalysts into the game. The development of new catalysts for the selective oxidation of *iso*butane is at the intersection between strong economic interests for optimizing the synthesis of chemical building blocks and the enduring scientific challenge to activate this very inert compounds for functionalization [34].

In this work, the catalysts used in selective oxidation of *iso*butane to MAA and MAC in recent years and their properties are reviewed, and the reaction conditions that play a decisive role in this reaction are comprehensively and systematically analyzed and discussed.

## 2. Catalysts for Selective Oxidation of *Iso*butane to MAA and MAC

As aforementioned, the acidic and redox properties of the catalyst play crucial roles in the selective oxidation of *iso*butane to MAA and MAC, as they contribute to activate the C–H bond of *iso*butane to generate the desired oxidized products. Thus, in this section, all kinds of catalysts that have been reported so far in the literature for the selective oxidation of *iso*butane to MAA and MAC will be systematically summed up.

### 2.1. Keggin-Type Heteropolycompounds Catalysts

To date, Keggin-type HPAs, which present superior catalytic activity for the selective oxidation of *iso*butane to MAA and MAC, have been proposed to be the most promising catalysts. The acidic and redox properties of Keggin-type HPAs can be easily tuned by substituting appropriate elements into the catalyst structure.

#### 2.1.1. Keggin Structure

Before discussing the efficient Keggin-type HPAs catalysts for this reaction, it is necessary to outline the main characteristics of the structure of the Keggin-type HPAs. HPAs, also known as polyoxometalates (POMs), are nanoscale inorganic metal-oxygen clusters formed by oxygen bonding of pretransition metals (Mo, W, V, Nb, Ta, etc.) [38]. In 1934, Keggin used X-ray powder diffraction to determine the famous eponym Keggin

structure model for the first time [39,40]. After this, Dawson, Anderson, Waugh, Silverton, and Lindqvist and other structures were determined successively. The complex structure of these compounds can be divided into two levels of organization [41]. The primary structure is the Keggin polyanion, consisting of the atomic arrangement of the heteropolyanion itself, and the general formula is $[XM_{12}O_{40}]^{n-}$ ($X$ = P, Si, Ge, As..., $M$ = Mo, W), where $X$ is the central atom, and where a central $XO_4$ tetrahedron is linked by the corners with 12 $MO_6$ octahedra, while $M$ is the coordination atom. Three octahedra are connected to form a trihedral $M_3O_{13}$ ensemble, whereby the octahedra within the same group are linked by the edges. In heteropoly acids (acid form) in the solid state, protons play an essential role in the structure of the crystal, by linking the neighboring heteropolyanions. The secondary structure can thus be depicted as a tridimensional arrangement between the heteropolyanions, the cations, and the crystallization water. The observed complex molecular organization is maintained by electrostatic forces or weak Van-der-Waals or hydrogen bounds [42]. Hence, it is flexible to different extents depending on the counter cation and the structure of the polyanion, and is the basis of bulk-type catalysis of solid HPAs. The tertiary structure is the secondary structure assembled into solid particles describing the physical characteristics of the material, such as the particle size, surface area, and pore structure, playing an important role in heterogeneous catalysis. Using Keggin-type heteropolyanion as an illustration, the primary, secondary, and tertiary structures of heteropolycompounds (HPCs) are depicted in Figure 1.

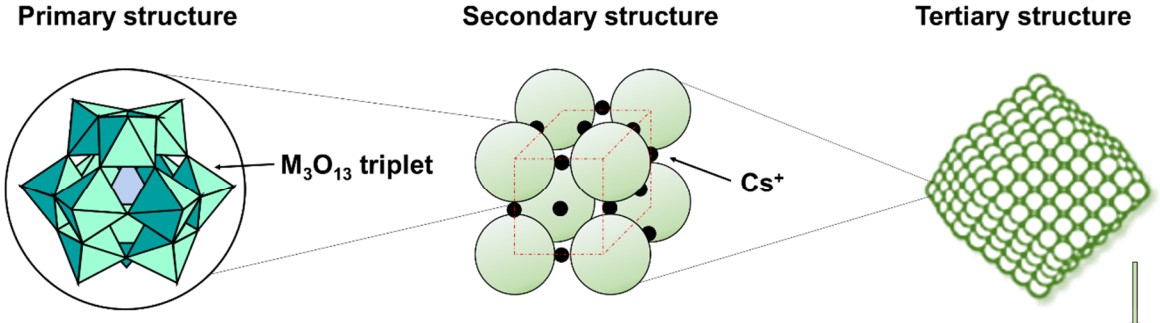

**Figure 1.** Hierarchical structure of HPAs in the solid state, as suggested in [1].

Based on the above composition and structural characteristics, Keggin-type HPAs usually exhibit two properties: Brønsted acidity and redox properties [40]. The strong Brønsted acidity is related to its highly symmetrical spatial structure. Due to the symmetry, cage structure, and nanometer size, the negative charge is highly delocalized in the Keggin anion, so that the charge density on the surface is low, the binding ability to the proton is weak, and consequently the proton activity is quite high [43,44]. The coordination atoms in Keggin-type HPAs are generally in their highest oxidation state. So, they can be used as multi-electron acceptors with certain oxidation properties. Notably, Mo-based heteropoly-acids accept electrons and the formation of reduced heteropolymolybdate anions, due to the blue color, named as "*heteropoly blue*". A common characteristic of heteropolyanions is their high reducibility. Electrochemical studies by Okuhara et al. [45] have shown that Keggin-type heteropolyanions revealed a series of reversible one- or two-electron reduction steps in aqueous or non-aqueous solutions, which produced a dark-colored mixed-valence species "*heteropoly blues*".

Nevertheless, the initial anion structure is kept after the heteropoly acid is reduced to "*heteropoly blue*". Heteropolyacid can be reoxidized to recover their initial oxidation state, hence showing a reversible redox property [40].

### 2.1.2. Heteropolycompounds for Oxidizing Isobutane to MAA and MAC

One of the first attempts at using Keggin-type HPA for selective oxidation of *iso*butane to MAA and MAC was performed with neat phosphomolybdic acid ($H_3PMo_{12}O_{40}$). However, the catalytic activity of this kind of catalyst was not satisfactory in terms of its low *iso*butane conversion (4–5%) and low MAA and MAC selectivity (10–16%). The poor stability and low surface area of $H_3PMo_{12}O_{40}$ (10 m$^2$/g) may be the probable reasons [30]. Li et al. [46] increased the catalytic performance of the $H_3PMo_{12}O_{40}$ catalyst by increasing its reducibility with pyridinium incorporation. The pyridinium $H_3PMo_{12}O_{40}$ catalyst was activated by heat treatment under a nitrogen stream up to 420 °C, whereby it presented a highly reduced state as a consequence. Around a 62% selectivity to MAA and MAC was achieved at 12% conversion of *iso*butane at a 300 °C reaction temperature. It was shown that two thirds of the pyridinium ion were eliminated during the heat treatment, and this led to oxygen-deficient heteropoly anions of reduced molybdenum. The oxygen-deficient anions provided sites for activating molecular oxygen and *iso*butane.

Cesium or/and Ammonium-Based HPCs

In order to adjust the acidity and redox properties of the catalyst, cation substitution of $H_3PMo_{12}O_{40}$ with cesium, ammonium, vanadium, and other metal ions was studied to improve the catalytic performance. It was found that inserting large counter-cations $Cs^+$ or/and $NH_4^+$ with $H^+$ in the secondary structure of $H_3PMo_{12}O_{40}$ improved the catalytic activity. In fact, this substitution has profound effects on the tertiary structure, resulting in the formation of micro- and mesopores, whereby the specific surface area increases. Mizuno et al. [47,48] proved that the catalytic properties were greatly enhanced by substitution of $Cs^+$ for $H^+$ in $H_3PMo_{12}O_{40}$. The conversion of *iso*butane and the selectivity to MAA and MAC improved by varying the amount of $Cs^+$ ($Cs_xH_{3-x}PMo_{12}O_{40}$, $x$ = 0–3), whereby the conversion and selectivity reached 16% and 31%, respectively, when the $Cs^+$ content was 2.5. Cavani et al. [28,33,49,50] prepared a $NH_4^+$-substituted catalyst, $(NH_4)_3PMo_{12}O_{40}$, whereby selectivity to MAA + MAC reached up to 54% at 8% *iso*butane conversion (350 °C reaction temperature). It is noteworthy that the catalytic activity of $(NH_4)_3PMo_{12}O_{40}$ was affected by the pH during the preparation process. As a matter of fact, both conversion and selectivity increased with pH (pH in a range between 0.5 and 4.0). This was explained by the fact that the $Mo^{n+}$ species migrated to the cationic position, thus forming a mixed salt of $Mo^{n+}(NH_4)_{7-n}PMo_{11}O_{40}$ during the preparation process. These $Mo^{n+}$ species allowed activation of the *iso*butane and transformation of *iso*butane into MAA. High pH produced more cationic $Mo^{n+}$-species, resulting in better catalytic activity in terms of *iso*butane conversion and MAA and MAC selectivity.

Some researchers have also prepared catalysts with mixed $Cs^+$ and $NH_4^+$ substitution of the proton in Keggin-type HPAs by varying the ammonia/cesium ratios. Jing et al. [51] synthesized a series of $Cs_x(NH_4)_{3-x}HPMo_{11}VO_{40}$ catalysts by varying the ammonia/cesium ratios, and investigated their surface area and acidity to correlate their properties to the catalytic activity. The results showed that increasing the ammonia content in the catalyst (decreasing the Cs content) decreases the specific surface area, whereas increasing the ammonia content increases the number of acid sites, as shown in Figure 2. This opposite trend allowed a large range of surface acid density to be covered, thus studying the influence of the latter in the selective oxidation of *iso*butane. As a matter of fact, the selective oxidation of isobutane over heteropolycompounds catalysts was classified as a surface-type reaction. Thus, it was found that the catalyst $Cs_{1.7}(NH_4)_{1.3}HPMo_{11}VO_{40}$, which presented a proper balance between acidity (0.72 mmol $g_{cat.}^{-1}$) and specific surface area (16.4 m$^2$ g$^{-1}$), exhibited the highest catalytic activity (*iso*butane conversion 9.6%, MAC+MAA selectivity 57%). Sultan et al. [30] also prepared a series of mixed Keggin-type heteropolysalts with various ammonia/cesium ratios. $Cs_{1.75}(NH_4)_{1.25}HPMo_{11}VO_{40}$ catalyst presented the highest selectivity to MAA + MAC (61%) at 6% *iso*butane conversion. The roles of ammonia and cesium in the catalysts were identified in this work: The cesium atoms of the catalysts did not participate in the reaction of *iso*butane but formed an alkaline salt of HPA that

played a role of large surface area support. This enables an efficient dispersion of the active phase, which is a key parameter for high catalytic performance. On the other hand, ammonium ions are partially eliminated during thermal treatment and reaction, whereby it was proposed that this elimination led to a release of $NH_3$ that could partially reduce the solid and hence modify its redox properties.

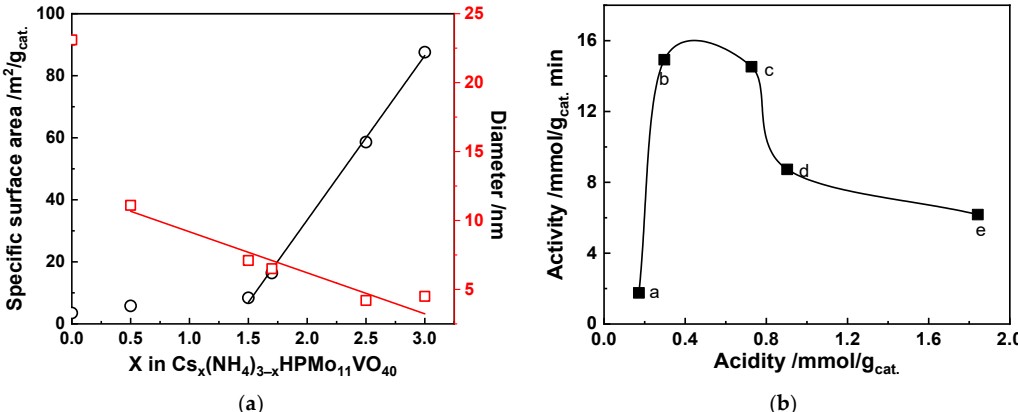

**Figure 2.** (**a**) Plots of surface area (circles) and pore diameter (squares) as a function of the Cs loading expressed as $x$ in the $Cs_x(NH_4)_{3-x}HPMo_{11}VO_{40}$ calcined samples, and (**b**) correlation between the catalytic activity and acidity [a: $Cs_3(NH_4)_0HPMo_{11}VO_{40}$, b: $Cs_{2.5}(NH_4)_{0.5}HPMo_{11}VO_{40}$, c: $Cs_{1.7}(NH_4)_{1.3}HPMo_{11}VO_{40}$, d: $Cs_{1.5}(NH_4)_{1.5}HPMo_{11}VO_{40}$, e: $Cs_{0.5}(NH_4)_{2.5}HPMo_{11}VO_{40}$]. Reaction conditions: 27 mol.% *iso*butane, 13.5 mol.% $O_2$, 10 mol.% $H_2O$, 49.5 mol.% He, temperature 340 °C, contact time 4.8 s. Drawn using data from [51].

Vanadium-Substituted HPCs

Partial substitution of Mo with vanadium ($V^{5+}$) in the Keggin primary structure also presents an important effect on the selective oxidation of *iso*butane to MAA and MAC. The highest oxidation state of vanadium ($V^{5+}$) increased the redox property of the catalyst by reducing $Mo^{6+}$ to $Mo^{5+}$ after the reaction owing to vanadium migration from the primary structure to the secondary one. The presence of V species in the catalyst may facilitate the reduction and re-oxidation process by promoting the transformation of adsorbed oxygen into lattice oxygen $O^{2-}$, which is more selective to MAA than electrophilic species, such as $O_2^-$ and $O^-$. Thus, substitution of Mo with vanadium ($V^{5+}$) increased the selectivity to MAA by accelerating the oxidation of MAC to MAA, and suppressed further oxidation of MAC and MAA to $CO_x$. This was shown by Mizuno et al. [52], who prepared a series of catalysts with various vanadium contents $Cs_{2.5}Ni_{0.08}H_{0.34+x}PV_xMo_{12-x}O_{40}$ ($x$ = 0–3). The conversion and the selectivity to MAA increased from $x$ = 0 to 1. Then, both conversion and selectivity to MAA decreased when further increasing the vanadium content to $x$ = 2 and 3, as shown in Table 1. The increase of selectivity to MAA by mono-$V^{5+}$-substitution may be explained by the acceleration of oxidation of MAC to MAA by vanadium ion, which exists as $VO_2^+$ and/or $VO^{2+}$. The decrease in selectivity to MAA with high vanadium content would relate to the formation of $VO_2$.

**Table 1.** Oxidation of *iso*butane catalyzed by $Cs_{2.5}Ni_{0.08}H_{0.34+x}PV_xMo_{12-x}O_{40}$ [52].

| $x$ | Conversion [a], % | Selectivity, % | | | |
|---|---|---|---|---|---|
| | | MAA | MAC | CO | $CO_2$ |
| 0 | 10 | 27 | 12 | 30 | 26 |
| 1 | 15 | 36 | 9 | 25 | 24 |
| 2 | 13 | 28 | 8 | 25 | 33 |
| 3 | 12 | 10 | 8 | 35 | 38 |

[a] Reaction conditions: 17 mol.% *iso*butane, 33 mol.% $O_2$, 50 mol.% $N_2$, temperature 320 °C, total flow rate 30 $cm^3 \cdot min^{-1}$.

In order to increase the amount of $VO^{2+}$ species at the surface of the catalyst, He et al. [53] recently prepared a series of $Cs_2V_xPMo_{11}VO_{40}$ ($x = 0$–0.5) catalyst with vanadyl species in both the Keggin unit and secondary structure. The results showed that the acid sites number increased with increasing the $VO^{2+}$ content at the surface. Around 59% selectivity to MAA and MAC was obtained at 9% of *iso*butane conversion for $Cs_2V_{0.3}PMo_{11}VO_{40}$ catalyst. Similarly, Liu et al. [27] added different amounts of $VO^{2+}$ in the secondary structure of the $Cs_2V_xCu_{0.2}PMo_{12}O_{40}$ catalyst. The selectivity to MAA+MAC increased to 53% when $x = 0.3$, which was much higher than for the non-promoted catalyst (with a selectivity to MAA+MAC of 21%), at an almost constant conversion of 13%.

Metal Ions-Substituted HPCs

Transition-metal-substituted heteropolyacids have also attracted much attention for the selective oxidation of *iso*butane to MAA and MAC due to their unique and remarkable catalytic performances. The effect of iron on the catalytic activity of cesium heteropolyacids salts has been deeply studied. It should be noted that the effect of iron on the catalytic activity depends on the positions of integration: Metal- or proton substitution. Min et al. [54] prepared proton-substituted $Cs_{2.5}Fe_xH_{0.5-3x}PMo_{12}O_{40}$ catalysts with iron content ($x$) from 0 to 0.16. The results showed that the addition of iron resulted in a slight decrease in *iso*butane conversion but benefited from the selectivity in MAA and MAC for $0 < x < 0.08$, while it decreased with higher iron content. Therefore, the highest yield for MAA and MAC was reached at an iron content of 0.08, as shown in Figure 3. The same trend was also obtained by Mizuno et al. [55]. The highest yield of MAA and MAC was 6.3% at 14% *iso*butane conversion over $Cs_{2.5}Fe_{0.08}H_{0.26}PMo_{12}O_{40}$. The increased selectivity to MAA and MAC over iron-substituted catalysts was due to the suppression of complete oxidation of MAA and MAC generated and the acceleration of the oxidation of MAC to MAA by iron.

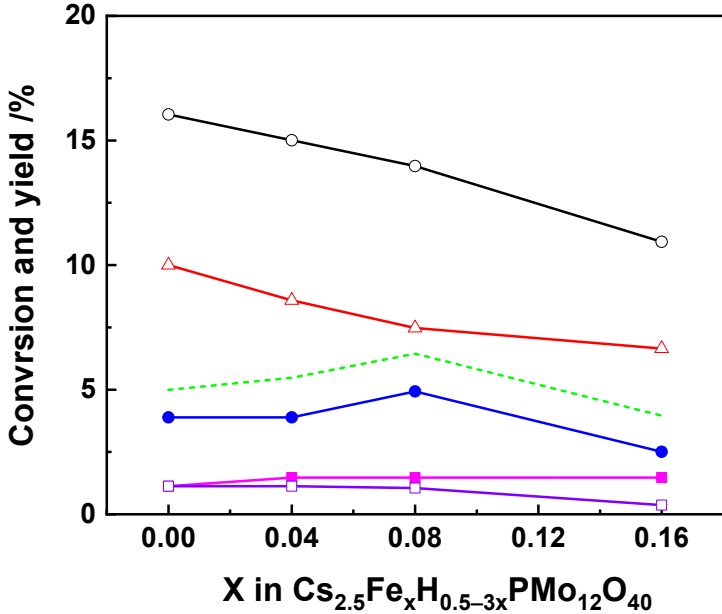

**Figure 3.** Oxidation of *iso*butane catalyzed by $Cs_{2.5}Fe_xH_{0.5-3x}PMo_{12}O_{40}$ at 340 °C. (○), (●), (■), (□), and (△) represent conversion of *iso*butane and yields of MAA, MAC, AA, and $CO_x$, respectively. The broken line indicates the sum of yields of methacrylic acid and methacrolein. Reaction conditions: 17 mol.% *iso*butane, 33 mol.% $O_2$, 50 mol.% $N_2$, temperature 340 °C, total flow rate 30 $cm^3 \cdot min^{-1}$. Drawn using data from [54].

A comparison of the catalysts with iron as a counter-cation or inserted inside the Keggin-anion was made by Knapp et al. [56]. The selectivity to MAC and MAA was 15% over $Cs_{1.5}(NH_4)_2PMo_{11.5}Fe_{0.5}O_{39.5}$ catalyst, where iron was inserted inside the Keggin-

anion, which was lower than that obtained over $Cs_{1.5}Fe_{0.5}(NH_4)_2PMo_{12}O_{40}$ catalyst (21%), where iron was substituted as a counter-cation. The conversion of *iso*butane over both catalysts at 360 °C was maintained at 8%. Thus, iron showed a negative effect on the catalytic activity when it was inserted into the Keggin-anion. Furthermore, iron can be released from the Keggin-anion, yielding iron counter-cations and a highly unstable lacunary $PMo_{11}O_{39}{}^{7-}$ HPC, which was easily rearranged as $PMo_{12}O_{40}{}^{3-}$ under reaction conditions. Meanwhile, the released iron favored the elimination of $NH_4^+$, whereby the stability of the catalyst decreased. Hence, only the catalyst with iron in counter-cation positions presented a positive effect on the catalytic activity. The addition of iron as a counter-cation in the catalyst would increase the acidity due to the presence of hydrated iron cations $Fe(OH)_n{}^{(3-n)+}$ [57].

Next to Fe, the proton substitution with Ni was studied. A small amount of nickel addition to cesium heteropolyacids salts showed a positive effect on the catalytic activity. Mizuno et al. [48] prepared a $Cs_{2.5}Ni_xH_{0.5-2x}PMo_{12}O_{40}$ catalyst by varying the nickel content from 0 to 0.16. The conversion of *iso*butane and the yields for MAA and MAC increased to 24% and 8% at $x = 0.08$ from 16% and 5% at $x = 0$. However, the conversion and the yields of MAA and MAC decreased with further increasing of the nickel content over $x = 0.08$. It was proposed that the addition of a low quantity of Ni ($x < 0.08$) allowed acceleration of the rate-determining step of the reaction, i.e., the activation of *iso*butane.

The addition of copper to $Cs_2Cu_x{}^{2+}H_{1-2x}PMo_{12}O_{40}$ ($0 < x < 0.43$) catalyst was studied by Langpape et al. [58] The results showed that increasing the copper content reduced the conversion of isobutane and the selectivity to MAA from 7.2% to 5.9% and from 12% to 5%. The selectivity to MAC remained stable (around 15%). The decreased selectivity to MAA was explained by the increased acidity of the catalyst with increasing of the copper content, leading to the degradation of MAA by a consecutive oxidation reaction. However, recently, Liu et al. [27] found that co-incorporation of $Cu^{2+}$ and $VO^{2+}$ in the secondary structure in the $Cs_2V_{0.3}Cu_{0.2}PMo_{12}O_{40}$ catalyst presented a higher catalytic activity (53% selectivity to MAA and MAC, 13% *iso*butane conversion) than that without adding $Cu^{2+}$ and $VO^{2+}$ (23% selectivity to MAA and MAC, 12% isobutane conversion). This was explained by a synergetic effect between $Cu^{2+}$ and $VO^{2+}$. The synergetic effect was related to the charge transfer between $Mo^{6+}$ in the Keggin anion and $Cu^{2+}$ counter cation, which promoted the re-oxidation of the catalyst ($Cu^{2+} + Mo^{5+} \rightarrow Mo^{6+} + Cu^+$). In addition, the effect of different transition metal substitution ($Cu^{2+}$, $Fe^{3+}$, $Ni^{2+}$, and $Ce^{4+}$) on the catalytic activity was also investigated, as shown in Figure 4. The results showed that the $Cs_2V_{0.3}Cu_{0.2}PMo_{12}O_{40}$ catalyst with co-incorporation of $Cu^{2+}$ and $VO^{2+}$ in its secondary structure exhibited superior catalytic activity and selectivity to MAA for the oxidation of isobutane compared to that with co-doped $Fe^{3+}$, $Ni^{2+}$, and $Ce^{4+}$. It was proposed that the reduction of $Cu^{2+}$ ions (~350 °C) located in the secondary structure was easier than the reduction of $Fe^{3+}$, $Ni^{2+}$, and $Ce^{4+}$ (>600 °C), as determined by $H_2$-TPR profiles. The latter facilitated the redox process of the catalyst under reducing conditions for $Cu^{2+}$-containing catalyst, which accelerated the electron transfer between the Cu and Mo cations and increased the supply of lattice oxygen for the re-oxidation process.

Wu et al. [59] prepared a Keggin-type molybdovanadophosphoric heteropoly acid catalyst with protons completely substituted by tellurium. The catalytic results showed that the selectivity to MAC increased remarkably over $Te_2PMo_{11}VO_n$ catalyst (22%) compared with that over $H_4PMo_{11}VO_{40}$ (6%). However, the conversion of *iso*butane and the selectivity to MAA decreased from 6.2% to 5.5% and from 30% to 27%, respectively. The increase in the selectivity to MAC over the Te-substituted heteropolycompound catalyst was explained by the abstraction of $\alpha$-H from *iso*butene intermediate by tellurium. After, a higher selectivity to MAC and MAA was obtained by mechanically mixing 70 wt.% of a phosphomolybdic cesium salt containing Te as counter-cation and 30 wt.% of $\beta$-$La_2Mo_{1.9}V_{0.1}O_{8.95}$ (LMV) (61%) in comparison with the Te-substituted HPCs (57%). The conversion of *iso*butane was kept at 19% for both catalysts at a 369 °C reaction temperature. It was thus proposed that since LMV is much larger than HPCs, HPCs can be uniformly adsorbed on LMV within

the first hour of the reaction, the support of the LMV stabilizes the HPCs and prevents the sintering of the active phase, and the synergy between the two phases results in good activity [60].

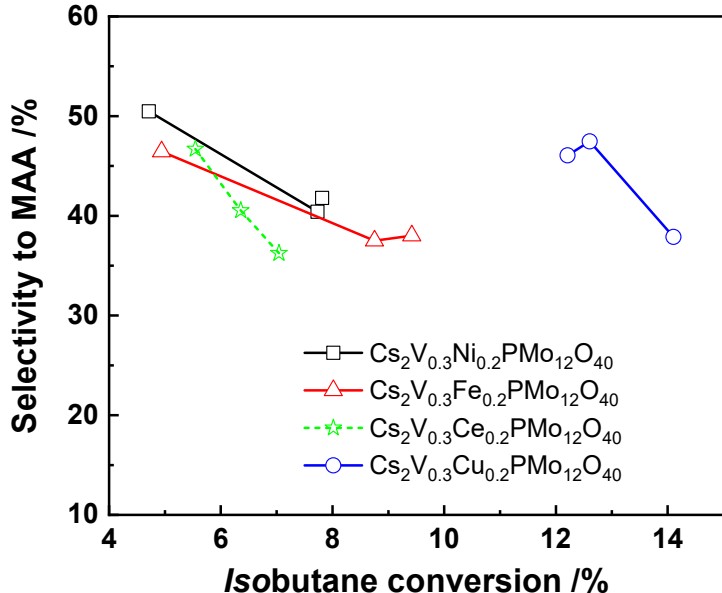

**Figure 4.** Selectivity to MAA as a function of *iso*butane conversion over V-containing Cs salt of heteropoly compounds with the incorporation of different transition metals. Reaction conditions: 27 mol.% *iso*butane, 13.5 mol.% $O_2$, 10 mol.% $H_2O$, 49.5 mol.% $N_2$, temperature 330 °C, total flow rate 20 cm³·min⁻¹. Drawn using data from [27].

The addition of antimony to HPCs also turned out to present high selectivity to MAA. Cavani et al. [50] showed that $(NH_4)_3PMo_{12}O_{40}$ doped with $Sb^{3+}$ ion presented 45% selectivity to MAA at 11% *iso*butane conversion, which was much higher than for an undoped catalyst (5% selectivity to MAA, 6% *iso*butane conversion) at 350 °C under *iso*butane lean reaction conditions (1% *iso*butane; 13% oxygen). The considerable improvement of selectivity was ascribed to the acceleration of electron exchange between $Sb^{3+}$ and $Mo^{6+}$ ($Sb^{3+} + 2Mo^{6+} \rightarrow Sb^{5+} + 2Mo^{5+}$), which decreased the oxidation state of molybdenum. However, the increase in MAA selectivity was much lower (from 43% to 50%) under *iso*butane-rich conditions (26% *iso*butane; 13% oxygen). This is because the undoped catalyst reduced by *iso*butane is more selective to MAA, whereby $Sb^{3+}$ was no longer necessary to decrease the oxidation state of molybdenum.

The catalytic performances of the reported Keggin-type HPCs-based catalysts related to the selective oxidation of *iso*butane to MAA and MAC are summarized in Table 2.

**Table 2.** Reported catalytic performance over different Keggin-type heteropolycompounds-based catalysts in the oxidation of isobutane.

| Catalyst | Reaction *T*, °C | Conversion, % | Selectivity, % | | | Yield, % (MAA + MAC) | Ref. |
|---|---|---|---|---|---|---|---|
| | | | MAA | MAC | CO_x | | |
| $H_3PMo_{12}O_{40}$ | 340 | 5 | 3 | 7 | 82 | 0.5 | [61] |
| $H_3PMo_{12}O_{40}$ | 340 | 4 | 4 | 12 | - | 0.6 | [30] |
| $H_3PMo_{12}O_{40}$ | 300 | 12 | 58 | 4 | 22 | 7.4 | [46] |
| $Cs_2HPMo_{12}O_{40}$ | 340 | 11 | 34 | 10 | 50 | 4.8 | [47] |
| $Cs_2HPMo_{12}O_{40}$ | 350 | 9 | 44 | 7.3 | 33 | 4.6 | [53] |
| $Cs_{2.5}H_{0.5}PMo_{12}O_{40}$ | 340 | 16 | 24 | 7 | 62 | 5.1 | [48] |
| $(NH_4)_3PMo_{12}O_{40}$ | 340 | 4 | 33 | 21 | - | 2.2 | [30] |
| $(NH_4)_3PMo_{12}O_{40}$ | 350 | 7 | 42 | 7 | - | 3.4 | [28] |

**Table 2.** *Cont.*

| Catalyst | Reaction $T$, °C | Conversion, % | Selectivity, % | | | Yield, % (MAA + MAC) | Ref. |
|---|---|---|---|---|---|---|---|
| | | | **MAA** | **MAC** | **$CO_x$** | | |
| $(NH_4)_3PMo_{12}O_{40}$ | 350 | 8 | 41 | 7 | - | 3.8 | [33] |
| $(NH_4)_3PMo_{12}O_{40}$ | 380 | 8 | 41 | 13 | 37 | 4.3 | [49] |
| $(NH_4)_3PMo_{12}O_{40}$ | 350 | 6 | 43 | 12 | 36 | 3.3 | [50] |
| $Cs_{0.5}(NH_4)_{2.5}HPMo_{11}VO_{40}$ | 340 | 4 | 41 | 29 | 17 | 2.9 | [51] |
| $Cs_{1.15}(NH_4)_{1.85}HPMo_{11}VO_{40}$ | 340 | 6 | 45 | 15 | - | 3.6 | [30] |
| $Cs_{1.5}(NH_4)_{1.5}HPMo_{11}VO_{40}$ | 340 | 6 | 45 | 22 | 19 | 3.9 | [51] |
| $Cs_{1.7}(NH_4)_{1.3}HPMo_{11}VO_{40}$ | 340 | 10 | 44 | 14 | 26 | 5.5 | [51] |
| $Cs_{1.75}(NH_4)_{1.25}HPMo_{11}VO_{40}$ | 340 | 10 | 32 | 8 | - | 4 | [30] |
| $Cs_{2.4}(NH_4)_{0.6}HPMo_{11}VO_{40}$ | 340 | 2 | 10 | 16 | - | 0.5 | [30] |
| $Cs_{2.5}(NH_4)_{0.5}HPMo_{11}VO_{40}$ | 340 | 10 | 19 | 10 | 53 | 2.9 | [51] |
| $H_4PMo_{11}VO_{40}$ | 340 | 3 | 25 | 39 | - | 1.9 | [30] |
| $H_5PMo_{10}V_2O_{40}$ | 340 | 5 | 34 | 28 | 36 | 3 | [52] |
| $Cs_{1.6}H_{2.4}P_{1.7}Mo_{11}V_{1.1}O_{40}$ | 350 | 11 | 38 | 8 | 15 | 4.8 | [62] |
| $(NH_4)_3HPMo_{11}VO_{40}$ | 340 | 2 | 49 | 32 | - | 1.6 | [30] |
| $(NH_4)_4PMo_{11}VO_{40}$ | 350 | 3 | 52 | 20 | - | 2.2 | [28] |
| $Cs_2V_{0.3}PMo_{11}VO_{40}$ | 350 | 6 | 55 | 10 | 23 | 4 | [53] |
| $Cs_2V_{0.3}Cu_{0.2}PMo_{12}O_{40}$ | 350 | 13 | 48 | 6 | 31 | 7 | [27] |
| $Cs_{2.5}Fe_{0.08}H_{0.26}PMo_{12}O_{40}$ | 340 | 15 | | 30 | 40 | 4.5 | [54] |
| $Cs_{2.5}Fe_{0.08}H_{1.26}PMo_{12}O_{40}$ | 340 | 14 | 35 | 11 | 53 | 6.3 | [55] |
| $Cs_2V_{0.2}Fe_{0.2}PMo_{12}O_{40}$ | 350 | 9 | 38 | 9 | 39 | 4.2 | [27] |
| $Cs_2Fe_{0.2}H_{0.4}PMo_{12}O_{40}$ | 340 | 7 | 24 | 17 | 50 | 2.9 | [58] |
| $Cs_2Fe_{0.2}H_{0.4}PMo_{12}O_{40}$ | 340 | 7 | 24 | 17 | 46 | 2.9 | [61] |
| $Cs_2(NH_4)_xFe_{0.2}PMo_{12}O_{40}$ | 360 | 20 | 35 | 3 | 27 | 7.6 | [63] |
| $K_1Fe_{0.5}(NH_4)_{1.5}PMo_{12}O_{40}$ | 400 | 4 | 30 | 19 | 17 | 2 | [57] |
| $Cs_{1.5}Fe_{0.5}(NH_4)_2PMo_{12}O_{40}$ | 350 | 8 | 21 | 6 | - | 2.4 | [56] |
| $Cs_2Ni_{0.08}H_{1.34}PVMo_{11}O_{40}$ | 320 | 15 | 36 | 9 | - | 6.8 | [64] |
| $Cs_2V_{0.3}Ni_{0.2}PMo_{12}O_{40}$ | 350 | 8 | 41 | 9 | 34 | 4 | [27] |
| $Cs_{2.5}Cu_{0.08}H_{1.34}PVMo_{11}O_{40}$ | 340 | 14 | - | 35 | 36 | 4.9 | [65] |
| $Cs_2Cu_{0.2}H_{0.6}PMo_{12}O_{40}$ | 340 | 8 | 6 | 15 | 72 | 1.7 | [58] |
| $Cs_2V_{0.3}Cu_{0.2}PMo_{12}O_{40}$ | 350 | 13 | 48 | 6 | 31 | 7 | [27] |
| $Cs_2V_{0.2}Ce_{0.2}PMo_{12}O_{40}$ | 350 | 6 | 41 | 12 | 34 | 3.2 | [27] |
| $Cs_2Te_{0.2}H_xPMo_{12}O_{40}$ | 330 | 7 | 60 | 16 | 18 | 5.3 | [66] |
| $Cs_2Te_{0.3}(VO)_{0.1}H_xPMo_{12}O_{40}$ | 360 | 16 | 71 | 3 | 19 | 11.8 | [60] |
| $Te_2PMo_{11}O_n$ | 350 | 10 | 27 | 22 | 42 | 4.9 | [59] |
| $Te_{2.25}PMo_9V_3O_n$ | 390 | 12 | 22 | 34 | 36 | 6.7 | [67] |
| $(NH_4)_3PMo_{12}O_{40}/Sb_{0.23}O_x$ | 350 | 8 | 50 | 10 | 30 | 4.8 | [50] |
| $Mo_{12}V_{0.5}P_{1.5}As_{0.4}Cu_{0.3}Cs_{1.4}O_x$ | 370 | 0.4 | 18 | 30 | - | 0.2 | [68] |

### 2.1.3. Supported Heteropolycompounds

The main drawback of HPC-based catalysts is their poor stability on stream, which will be discussed in Section 3.1, and also their low specific surface area. To overcome these shortcomings, it is possible to synthesize supported catalysts. Because the selective oxidation of *iso*butane is supposed to be a surface-type reaction, the surface area of the catalyst is critical to catalytic activity. To obtain highly dispersed HPCs catalysts, several supports have been mentioned for applications in the selective oxidation of *iso*butane in the literature. Jing et al. [69] impregnated $(NH_4)_3HPMo_{11}VO_{40}$ (APMV) over different types of supports, such as commercial $SiO_2$, SBA-15, $ZrO_2$-grafted SBA-15, and $Cs_3PMo_{12}O_{40}$ (CPM). The results showed that the silica-supported catalysts did not lead to a good catalytic activity, even if they presented a high specific surface area. This was ascribed to the fact that APMV was decomposed under calcination and reaction conditions. The $NH_4^+$ and residual $H^+$ were progressively eliminated under the form of ammonia and water. The acidity and the activity drastically decreased as a consequence. The data in Table 3 show the acidic sites, specific surface area, and catalytic performance of the catalysts. In the case of APMV/CPM, the strong (49%) and very strong (36%) sites were predominant, which

is probably related to the high catalytic performance. Similar results were also obtained by Cai et al. [70], who supported 40 wt.% of APMV on $CeO_2$, $WO_3/ZrO_2$, and molecular sieves. It was found that the CPM support had the best chemical compatibility with the active phase, leading to a better stabilization ability of active phase APMV. Consequently, CPM-supported APMV catalyst presented the best catalytic activity (11% conversion of *iso*butane, 5.4% yield of MAA and MAC). Actually, the most important factor for the high catalytic activity of this supported catalyst was the increase of surface acid sites' density. The surface acid sites' density of CPM-supported catalyst was much higher than that of other supported catalysts, leading to superior catalytic activity, as shown in Figure 5.

**Table 3.** The amount of acidic sites and distribution of acid strengths, the specific surface area, and the catalytic performance in the oxidation of isobutane of the supported catalysts [69].

| Catalyst | Acidity, $NH_3$ Uptake, mol/m² cat (mmol/$g_{cat}$) | | | | | $S_{BET}$, (m²/g) | Conversion [a] (%) | Selectivity (mol.%) |
| | 130–300 °C, Weak | 300–450 °C, Medium | 450–560 °C, Strong | 560–700 °C, Very Strong | Total Acidity | | IBAN | MAC + MAA |
|---|---|---|---|---|---|---|---|---|
| APMV/CPM | 2.9 (0.05) | 12.4 (0.21) | 52.4 (0.89) | 38.2 (0.65) | 105.9 (1.80) | 17 | 15.3 | 52.0 |
| APMV/$SiO_2$ | 1.1 (0.17) | 0.9 (0.14) | 7.76 (1.23) | - | 9.7 (1.54) | 159 | 11.3 | 27.4 |
| APMV/$ZrO_2$/SBA-15 | 2.1 (0.6) | 2.2 (0.62) | - | - | 4.3 (1.22) | 287 | 6.6 | 11.2 |
| APMV/SBA-15 | 0.4 (0.16) | 1.0 (0.35) | 0.9 (0.32) | 0.1 (0.03) | 2.4 (0.87) | 368 | 5.0 | 27.7 |

[a] Reaction conditions: temperature 340 °C, atmospheric pressure, contact time 4.8 s, molar ratios: IBAN/$O_2$/$H_2O$/inert = 27/13.5/10/49.5.

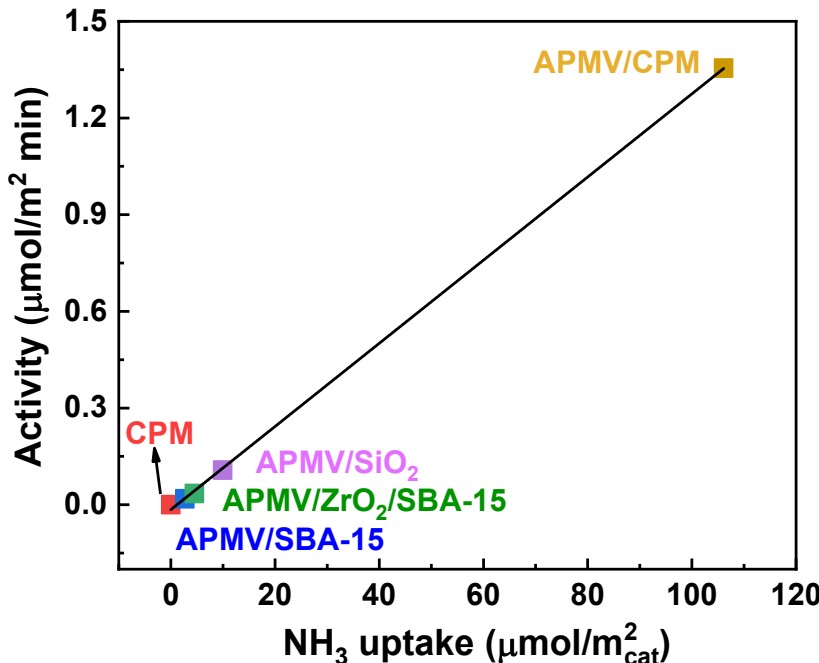

**Figure 5.** Correlation between the activity in isobutane oxidation and the surface acid site density of supported catalysts measured by $NH_3$-TPD. Reaction conditions: 27 mol.% *iso*butane, 13.5 mol.% $O_2$, 10 mol.% $H_2O$, 49.5 mol.% He, temperature 340 °C, contact time 4.8 s. Drawn using data from [69].

Then, the active phase loading of APMV on the CPM support has also been investigated, as the concentration of active phase affects the catalytic performance [26]. It was found that best catalytic performance in terms of *iso*butane conversion (16%), and MAA and MAC selectivity (52%) was obtained on 40 wt.% APMV/CPM catalyst. The surface acid sites' density, which played a crucial role in the catalytic activity, increased with increasing the loading amount. Meanwhile, the thermal stability of the active phase also increased with increasing the loading. The catalytic activity of the supported HPCs reported in the literature are summarized in Table 4.

**Table 4.** Catalytic performance in the oxidation of *iso*butane over supported HPCs catalysts.

| Catalysts | Reaction $T$, °C | Conversion, % | Selectivity % | | | Yield, % (MAA + MAC) | Ref. |
|---|---|---|---|---|---|---|---|
| | | | MAA | MAC | $CO_x$ | | |
| $(NH_4)_3HPMo_{11}VO_{40}/SBA-15$ | 340 | 5 | 10 | 18 | 42 | 30 | [69] |
| $(NH_4)_3HPMo_{11}VO_{40}/ZrO_2/SBA-15$ | 340 | 7 | 2 | 10 | 64 | 0.7 | [69] |
| $(NH_4)_3HPMo_{11}VO_{40}/SiO_2$ | 340 | 11 | 13 | 15 | 43 | 3 | [69] |
| $H_3PMo_{12}O_{40}/SiO_2$ | 350 | 7 | 40 | 9 | 39 | 3.4 | [71] |
| $(NH_4)_3HPMo_{11}VO_{40}/Molecular sieve$ | 340 | 8 | 0.6 | 0.9 | 80 | 0.1 | [70] |
| $(NH_4)_3HPMo_{11}VO_{40}/CeO_2$ | 340 | 8 | 0.3 | 0 | 99 | 0.02 | [70] |
| $(NH_4)_3HPMo_{11}VO_{40}/WO_3/ZrO_2$ | 340 | 7 | 0.9 | 4 | 85 | 0.3 | [70] |
| $(NH_4)_3HPMo_{11}VO_{40}/Cs_{2.5}H_{0.5}PMo_{12}O_{40}$ | 340 | 9 | 35 | 6 | 38 | 4.6 | [70] |
| $(NH_4)_3HPMo_{11}VO_{40}/Cs_3PMo_{12}O_{40}$ | 340 | 11 | 43 | 5 | 42 | 5.4 | [70] |
| $(NH_4)_3HPMo_{11}VO_{40}/Cs_3HPMo_{11}VO_{40}$ | 340 | 10 | 35 | 5 | 47 | 4 | [70] |
| $(NH_4)_3HPMo_{11}VO_{40}/Cs_4PMo_{11}VO_{40}$ | 340 | 8 | 32 | 7 | 37 | 3.1 | [70] |
| $(NH_4)_3HPMo_{11}VO_{40}/Cs_3PMo_{12}O_{40}$ | 340 | 15 | 42 | 10 | 31 | 8 | [69] |
| $H_4PVMo_{11}O_{40}/Cs_3HPVMo_{11}O_{40}$ | 340 | 5 | 42 | 17 | 32 | 3 | [72] |
| $H_4PVMo_{11}O_{40}/Cs_3PMo_{12}O_{40}$ | 340 | 11 | 24 | 7 | 54 | 3.4 | [72] |

### 2.1.4. Reaction Mechanism over Heteropolycompounds

The activity/selectivity for the synthesis of MAA from *iso*butane is related to the combination of high proton acidity, fast electron transfer, and electron delocalization ability. As far as the reaction mechanism of the selective oxidation of *iso*butane by Keggin-type HPCs catalysts for the synthesis of MAA and MAC, Busca et al. [73,74] proposed a mechanism over $K_1(NH_4)_2PMo_{12}O_{40}$ catalysts, as shown in Figure 6. In the first step, the catalyst dehydrogenates the reactant to form a C=C double bond. *Iso*butane can be activated by the catalyst at the tertiary carbon by oxidative activation of the weakest C-H bond: two electrons are transferred to the catalyst, which is consequently reduced. This is considered as the rate-determining step of the reaction. In the following steps, the produced alkoxide is converted to an allylic alkoxy species, which reacts to the common intermediate dioxyalkylidene species, where the primary carbon is connected to the catalyst surface via two C-O-Mo bridges. Starting from this step, two parallel reactions may occur: one for the formation of MAC and the other one for the formation of MAA.

The reaction network was also confirmed by Paul et al. [30,75] based on a kinetic model. The initial selectivity for MAA and MAC was above 80% by the oxidation of *iso*butane, and the selectivity to carbon oxides and C2–C3 oxygenated products was less than 20%. Whereas the initially formed MAC was unstable, it was easier to transform to MAA and other by products as the transformation rate was around 50 times higher than *iso*butane conversion. It suggested that MAC was an intermediate in the formation of MAA. MAA was very stable without further transformation.

In order to establish the kinetic model for the oxidation process over HPCs catalysts, Paul et al. [75] proposed that *iso*butane reacts with oxidized sites of catalyst, rather than oxygen in the gas phase, in oxidation catalysis. This model was known as the Mars and Van Krevelen model. It was based on the redox dynamics of the HPCs sites: the reduction of the oxygen carrier (catalyst) by *iso*butane takes place in the first step, and in the second step, the reduced carrier is re-oxidized and regenerated by oxygen. The rate-limiting step of the reaction is the activation of the *iso*butane on the oxidized surface of the catalyst. The state of the catalyst can approach complete oxidation, except at a very low oxygen concentration. So, this study suggests that an increase in productivity could be obtained by favoring hydrocarbon activation rather than oxygen incorporation.

**Figure 6.** Reaction mechanism for the oxidation of isobutane over Keggin-type HPCs proposed in [74].

### 2.2. Mixed Metal Oxides

Mixed metal oxides catalysts have also been used as effective catalysts in the selective formation of MAC and MAA from *iso*butane. Due to the importance of the redox properties of catalysts in selective oxidation of *iso*butane, V and Mo are always combined together in the catalyst as the redox reaction occurs: $V^{5+} + Mo^{5+} \rightarrow V^{4+} + Mo^{6+}$. Guan et al. [76] prepared a series of Mo–V–O catalysts by varying the V/Mo molar ratio. It was found that the catalyst with a V/Mo ratio of 0.3 achieved the best MAC selectivity (40.4%) at an *iso*butane conversion of 6.4%. There was almost no MAA formed using these catalysts.

The mechanism of the catalytic oxidation of *iso*butane to methacrolein and methacrylic acid includes the activation of the C–H bond and subsequent oxygen insertion. Therefore, in most of the *iso*butane selective oxidation catalysts, in addition to V and Mo, other elements are used as dopants. Doping with other metal elements, such as Te, Sb, and Ce, and/or non-metal elements, such as P, could also improve the catalytic activity and products' distribution effectively over Mo–V–O metal oxide.

The group of Guan et al. also doped $MoV_{0.3}$ catalyst with different amounts of Te [77]. The Te/Mo ratio presented a clear effect on the catalytic activity. The conversion of *iso*butane increased with increasing it up to Te/Mo $\leq$ 0.25 and then decreased with further increasing of the Te content. Ultimately, the $MoV_{0.3}Te_{0.25}$ catalyst showed an MAC selectivity of 44.2% at an *iso*butane conversion of 15.6%, which was higher than that without the addition of Te. Nevertheless, Te had no effect on the selectivity of MAA, as the selectivity to MAA remained at 4%. This result was also confirmed by other researchers [78–81]. Furthermore, Sb was identified as another element that can increase the selectivity to MAC and the conversion of *iso*butane. The maximum conversion (20%) and MAC selectivity (39%) were obtained over $MoV_{0.3}Te_{0.23}Sb_{0.5}$ (Sb/Mo ratio of 0.5) catalyst at 470 °C [82]. It should be noted that the selectivity of MAA was 6% over $MoV_{0.3}Te_{0.23}Sb_{0.5}$, which was lower

than that over $MoV_{0.3}Te_{0.23}$ (9% selectivity to MAA). Therefore, the addition of Sb had a rather negative effect on the selectivity of MAA [83,84]. The catalytic activity can also be improved by inserting Ce into Mo–V–Te oxide. A 33% selectivity to MAC was obtained at 20% of *iso*butane conversion over $MoV_{0.3}Te_{0.23}Ce_{0.2}$ (Ce/Mo ratio is 0.2) at 420 °C [85]. The good catalytic performance observed upon adding Ce may be explained by the redox cycle of Ce and Mo ($Ce^{3+} + Mo^{6+} \rightarrow Ce^{4+} + Mo^{5+}$), and also by the improved mobility of lattice oxygen in the catalyst due to the strong capability of Ce for storing oxygen. As generally observed, the composition of metal oxide mixed catalyst has an important influence on the catalytic performance.

Adding phosphorous to metal oxide was also studied by Guan et al. [86]. The addition of phosphorous may favor the formation of MAC and suppress the consecutive reaction to $CO_x$. A 48% selectivity to MAC and MAA was achieved at 10% of isobutane conversion over $MoV_{0.3}Te_{0.23}P_{0.3}$ (P/Mo ratio is 0.3) at 440 °C, which was higher than without P addition (35% selectivity to MAC + MAA, 10% isobutane conversion). The precise reaction mechanism over phosphorous-doped metal oxide catalyst was not proposed. Similarly, a 51% MAC and MAA selectivity with 10% of isobutane conversion was obtained over $MoV_{0.3}Te_{0.23}Sb_{0.1}P_{0.3}$ (P/Mo ratio of 0.3), which was again higher than that without P (43% selectivity to MAC + MAA, 6% isobutane conversion).

The dispersion of metal oxides on a carrier is another way to improve the catalytic activity. Sun et al. supported 3 wt.% of $MoV_{0.8}Te_{0.23}O_x$ on $SiO_2$ and SBA-3 [87,88]. The results showed that the conversion of *iso*butane (9.2% for 3 wt.% $MoV_{0.8}Te_{0.23}O_x/SiO_2$, 13.4% for 3 wt.% $MoV_{0.8}Te_{0.23}O_x/SBA$-3) and the selectivity of MAC (33.7% for 3 wt.% $MoV_{0.8}Te_{0.23}O_x/SiO_2$, 28% for 3 wt.% $MoV_{0.8}Te_{0.23}O_x/SBA$-3) were both increased compared to the bulk $MoV_{0.8}Te_{0.23}O_x$ (0.3% conversion, 19.3% selectivity to MAC) at 440 °C due to the good dispersion of the active phases on large surface areas of the supports. However, the selectivity to MAA was not improved as no MAA was detected. In addition, no MAA was found over $MoV_{0.3}Bi_{0.3}O/ALPO_4$, even though this catalyst showed a selectivity of 45.1% for MAC at 9.5% conversion.

Usually, the mixed metal oxides do not have a special structure like heteropolycompounds, which exhibit the Keggin structure. They rather exhibit mixed crystalline phases or surface amorphous phases. As far as catalytic products are concerned, mixed metal catalysts are more favorable for the yield of MAC and low quantities of MAA. As far as the preparation technique is concerned, mixed metal oxides are generally obtained by a high-temperature thermal treatment. Therefore, the prepared catalysts could exhibit good thermal stability. This is definitely an advantage compared with the heteropolycompound catalysts. Nevertheless, the selectivity for MAA on metal oxides catalysts is very low or even undetectable, which is a major disadvantage compared to heteropolycompound catalysts. The catalytic activity of metal oxides reported in the literature is summarized in Table 5.

**Table 5.** Catalytic performance in the oxidation of *iso*butane over metal oxide catalysts.

| Catalysts | Reaction *T*., °C | Conversion, % | Selectivity % | | | Yields, % (MAA + MAC) | Ref. |
|---|---|---|---|---|---|---|---|
| | | | MAA | MAC | $CO_x$ | | |
| $MoV_{0.3}$ | 420 | 6.4 | 3.1 | 40.4 | 35 | 2.8 | [76] |
| $MoV_{0.3}Te_{0.25}$ | 420 | 15.6 | 3.5 | 44.2 | 25 | 7.4 | [77] |
| $MoV_{0.3}Te_{0.23}$ | 440 | 21.3 | 16 | 17 | 33 | 7.1 | [78] |
| $MoV_{0.3}Te_{0.25}$ | 420 | 20.1 | 11.7 | 12.2 | 59 | 4.8 | [79] |
| TeMoO | 380 | 20 | 3.8 | 33.5 | 26 | 7.5 | [80] |
| MoVTePO | 380 | 12.7 | 10.9 | 37 | 37 | 6.2 | [81] |
| $MoV_{0.3}Te_{0.23}Sb_{0.5}$ | 470 | 20 | 5 | 39 | 26 | 8.8 | [82] |
| $Mo_{12}V_3Sb_{85}O_x$ | 470 | 4.5 | - | 30 | 69 | 1.4 | [83] |
| $MoV_1Sb_{10}O_x$ | 440 | 8.1 | - | 17.6 | 59 | 1.4 | [84] |
| $MoV_{0.3}Te_{0.23}Ce_{0.2}$ | 420 | 20.2 | 20 | 33 | 28 | 10.7 | [85] |
| $MoV_{0.3}Te_{0.23}P_{0.3}$ | 440 | 9.4 | 15 | 33 | 33 | 4.5 | [86] |
| $MoV_{0.8}Te_{0.23}O_x/SiO_2$ | 440 | 9.2 | - | 33.7 | 54 | 3.1 | [87] |
| $MoV_{0.8}Te_{0.23}O_x/SBA$-3 | 440 | 13.4 | - | 28 | 56 | 3.7 | [87] |
| $MoV_{0.3}Bi_{0.3}O/AlPO_4$ | 540 | 9.5 | - | 45.1 | 43 | 4.3 | [88] |

## 3. Reaction Conditions of Selective Oxidation of *Iso*butane to MAA and MAC

The reaction conditions, such as feed composition, reaction temperature, and contact time, are the main factors that affect the catalytic performance. The reaction pressure will not be considered, as the reaction is always carried out at atmospheric pressure. Note that the ratio of hydrocarbons and oxygen has a significant effect.

### 3.1. Effect of the Feed Composition

For the selective oxidation of *iso*butane to MAA and MAC, the feed composition adopted by various companies [28,89,90] uses *iso*butane-rich conditions. This means *iso*butane-to-oxygen molar ratios between 2 and 0.8, as shown in the triangular diagram (ratio between inert gas-*iso*butane-oxygen) in Figure 7, indicating the flammability region at room temperature.

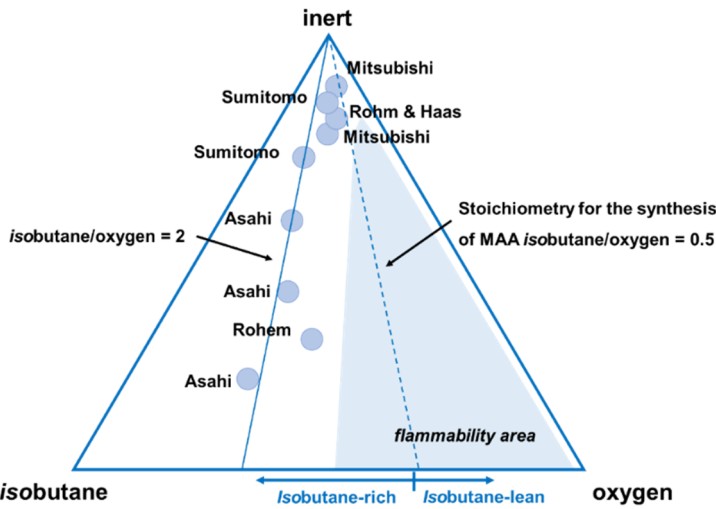

**Figure 7.** Ternary diagram of composition *iso*butane/oxygen/inert, showing the flammability area for mixtures at room temperature, and the feed composition claimed by several industrial companies. Reprinted with permission from Ref. [28]. Copyright 2021 Elsevier.

Cavani et al. [50] performed a deep investigation of the selective oxidation of *iso*butane under *iso*butane-lean and *iso*butane-rich conditions over a $(NH_4)_3PMo_{12}O_{40}$ catalyst, as shown in Figure 8. The selectivity to MAA was very low under *iso*butane-lean condition, and it was dramatically increased under the *iso*butane-rich condition. However, the selectivity to MAC was rather decreased under the *iso*butane-rich condition in comparison with that observed under the *iso*butane-lean condition. There was no big difference in the conversion under both conditions. This confirms that the *iso*butane-rich condition is preferable.

A lot of studies indicated that MAC and MAA selectivity was higher for a *iso*butane/$O_2$ molar ratio larger than 2 [21]. Huynh et al. [66] studied this parameter in the 1.4 to 3.5 range over $Cs_2Te_{0.2}V_{0.1}H_xPMo_{12}O_{40}$. The selectivity to MAA increased as the *iso*butane/$O_2$ ratio increased, implying that a high *iso*butane concentration favors the formation of MAA. However, the conversion also decreased with increasing of the *iso*butane/$O_2$ ratio.

At the same time, besides *iso*butane and oxygen in the feed composition, steam in the feed also has a positive effect [28,66,89], as shown in Figure 9. Due to the low *iso*butane conversions achieved, the recirculation of unconverted *iso*butane becomes a compulsory choice. Thus, steam decreases the concentration of *iso*butane and oxygen in the recycle loop and thus keeps the reactant mixture outside the flammability region. Furthermore, water also plays a positive role in the stability of HPC-based catalysts. In the case of heteropolyacids or salts, the presence of water favors the surface reconstruction of the Keggin structure, which decomposes during the reaction at high temperature, and the stability depends on the presence of $nH_2O$ in the secondary structure [34]. Additionally,

co-feed water also promotes desorption of methacrylic acid, preventing it from unselective consecutive reactions [90]. The *iso*butane conversion presented a maximum at a water content up to 10 mol.% and then decreased (Figure 9), most probably due to the competition of water and *iso*butane adsorption on the active sites. At the same time, a positive effect on the selectivity to MAA and MAC was observed at a low water content (≤10%), and the yield of MAA and MAC was optimal at the 10% water content [66].

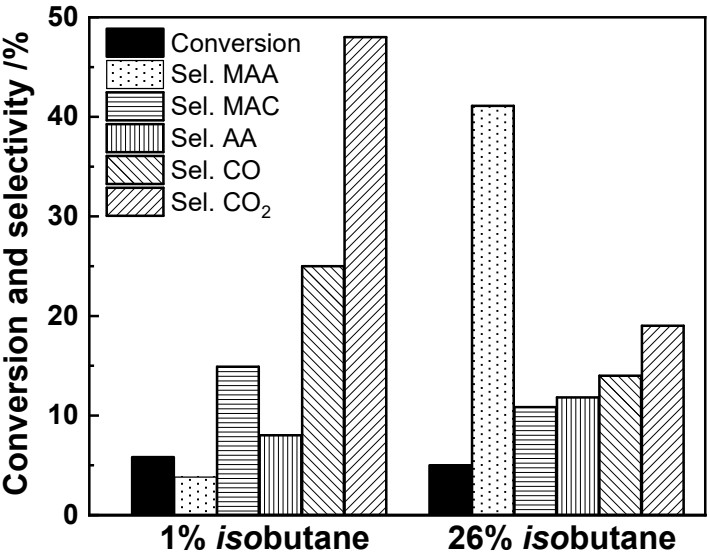

**Figure 8.** Comparison of the catalytic performance of an equilibrated $(NH_4)_3PMo_{12}O_{40}$ catalyst under *iso*butane-lean (1% *iso*butane, 13% oxygen; residence time 3.6 s; temperature 350 °C) and *iso*butane-rich (26% *iso*butane, 13% oxygen; residence time 3.6 s; temperature 352 °C). MAA: methacrylic acid; MAC: methacrolein; AA: acetic acid. Drawn using data from [50].

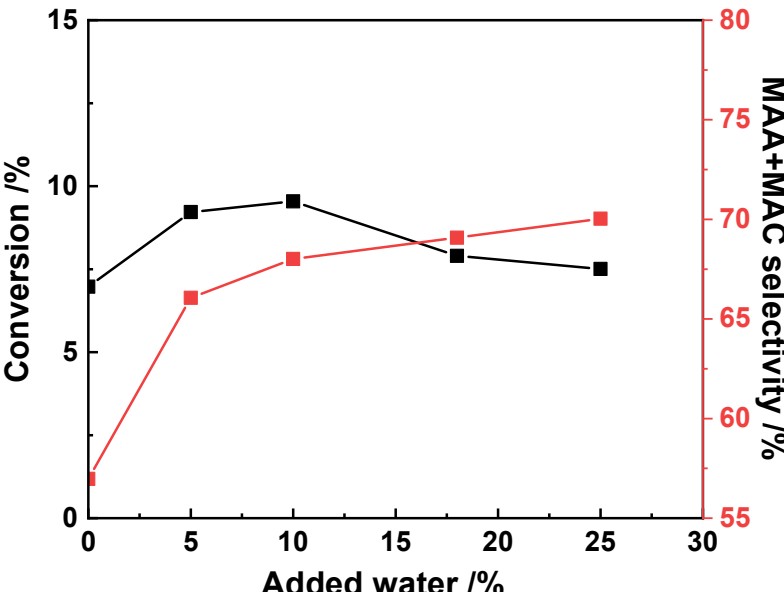

**Figure 9.** Effect of the addition of water on the *iso*butane conversion and on the selectivity to MAA and MAC. Reaction conditions: 27 mol.% *iso*butane, 13.5 mol.% $O_2$, $N_2$ balance, temperature 330 °C, contact time 4.8 s. Drawn using data from [66].

### 3.2. Effect of Reaction Temperature

Reaction temperature is a key factor that affects catalytic activity. The investigation of the influence of the reaction temperature on the catalytic performances in terms of *iso*butane

conversion and MAA and MAC selectivity were generally chosen in the range of 280 °C to 400 °C for Keggin-type heteropolycompounds-based catalysts according to the reported literature. In all cases, the conversion of *iso*butane increases with temperature, and the yield of MAA and MAC increases with temperature as well; however, the selectivity to MAA and MAC typically decreases with temperature due to the over-oxidation of the products. Jing et al. [26] studied the effect of the reaction temperature on the catalytic activity over $(NH_4)_3HPMo_{11}VO_{40}/Cs_3PMo_{12}O_{40}$ catalyst from 280 °C to 350 °C, as shown in Figure 10. The *iso*butane conversion increased linearly with increasing reaction temperature, whereby a maximum value was reached at 16%. Similarly, as temperature was increased, the yield of MAA and MAC increased. The selectivity to MAA was roughly constant with temperature, while a lower reaction temperature was more favorable for the formation of MAC, most probably since a high reaction temperature would result in the reaction of MAC to yield consecutive products (either selective or unselective). Cavani et al. [50] presented the same trend of the *iso*butane conversion and selectivity to MAA and MAC for a reaction temperature between 320 °C and 380 °C over $(NH_4)_3PMo_{12}O_{40}$. The conversion of *iso*butane increased from 1% to 6.5% with temperature, while the selectivity to MAC decreased from 20% to around 12%, with the selectivity to MAA constant at 45%.

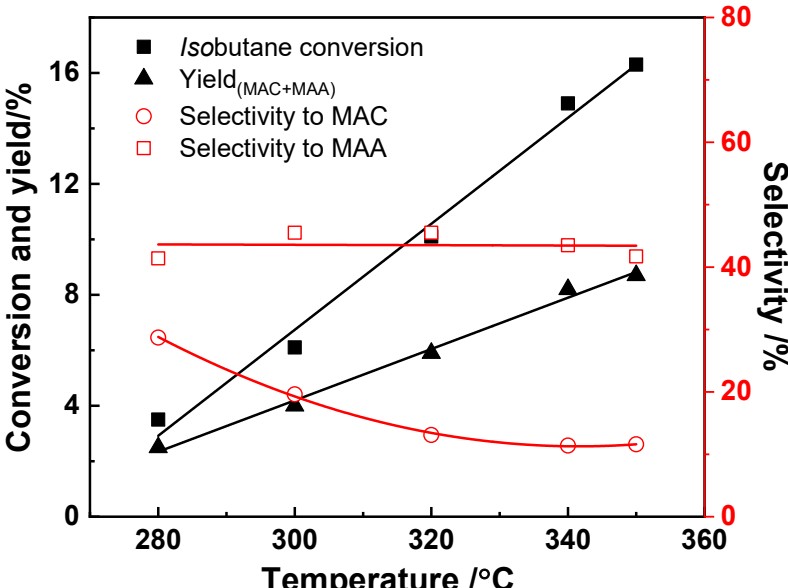

**Figure 10.** Effect of the reaction temperature on the catalytic performances. Reaction conditions: 27 mol.% *iso*butane, 13.5 mol.% $O_2$, 10 mol.% $H_2O$, 49.5 mol.% He, contact time 4.8 s. Drawn using data from [26].

Other researchers discovered a different trend for MAA selectivity by varying the reaction temperature. Liu et al. [63] showed that the selectivity to MAA increased with the temperature from 320 °C to 360 °C over $(NH_4)_xCsFe_{0.2}PMo_{12}O_{40}$, whereby a maximum was achieved at 360 °C, before it decreased at a higher reaction temperature (360 °C–400 °C). The selectivity to MAC decreased monotonically with temperature. The same trend was also found by Mizuno et al. [48], who showed that the selectivity to MAA presented a maximum value at 340 °C in the temperature range from 300 °C to 360 °C over $Cs_{2.5}Ni_{0.08}H_{0.34}PMo_{12}O_{40}$.

The selective oxidation of isobutane generally required a higher reaction temperature over mixed-metal oxides catalyst. Guan et al. [82] studied the catalytic performance with different reaction temperatures (360–480 °C) over $MoV_{0.3}Te_{0.23}Sb_{0.5}$ oxides. The results showed that the conversion of isobutane increased with temperature from 15% to 23%, while the selectivity to MAC increased dramatically from 12% at 360 °C to 39% at 470 °C, indicating that a high reaction temperature favored the formation of MAC, while a low

reaction temperature favored the formation of isobutene (50% selectivity to isobutene at 360 °C). The selectivity to MAA was relatively low with no more than 17% at 400 °C.

### 3.3. Effect of Contact Time

Next to the temperature and the feed composition, the contact time is another factor that can affect the catalytic activity and selectivity, based on the influence on the reaction kinetics of the complex reaction network. It was shown by Busca et al. [73] that the conversion of *iso*butane increased with increasing the contact time from 1 to 6 s at 350 °C over $K_1(NH_4)_2PMo_{12}O_{40}/Fe_1O_{1.5}$, suggesting that longer contact time was favorable for the activation of *iso*butane. However, the converted *iso*butane was more likely to form undesired products, such as AA and $CO_x$, rather than to form MAA and MAC as the selectivity to MAA and MAC was decreased while the selectivity to AA and $CO_x$ was increased after the longer contact time. For this result, they propose that the reaction network is composed of parallel reactions for the conversion of *iso*butane to MAA, MAC, and other by-products, and that the consecutive reactions for the oxidative degradation of MAA and MAC to AA, CO, and CO are responsible for the decrease of desired products (Scheme 1).

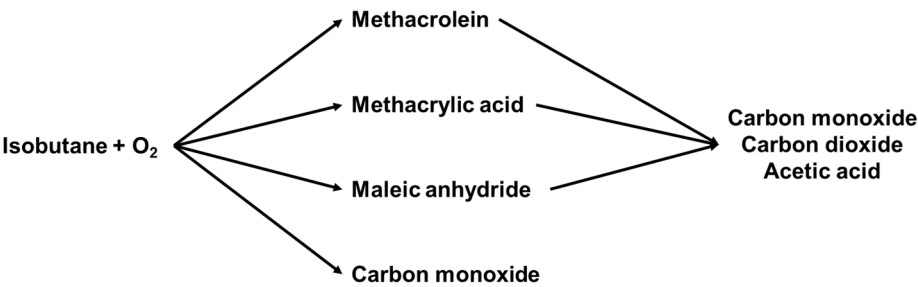

**Scheme 1.** Reaction network for the oxidation of *iso*butane at 350 °C over $K_1(NH_4)_2PMo_{12}O_{40}/Fe_1O_{1.5}$ as suggested in [73].

Sultan et al. [91] proposed a simplified reaction Scheme 2, based on a kinetic study with rate constant $K_3$ much greater than $K_1$, meaning that the desirable products react faster compared to *iso*butane, whereby high conversions become incompatible with high selectivity. In fact, both MAC and MAA are very reactive at high temperature, which is necessary for the activation of *iso*butane, and are thus readily transformed to degradation products [73]. Therefore, maintaining the high selectivity of the desired product requires that the temperature of the oxidation operation should be appropriately lowered.

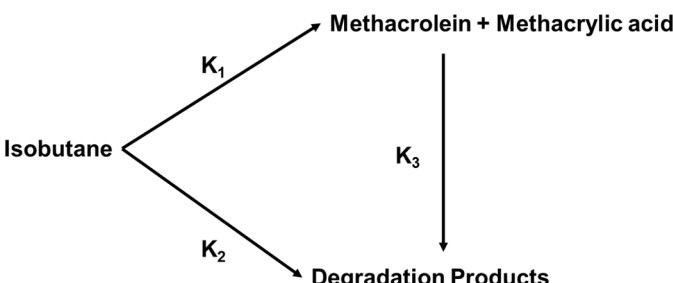

**Scheme 2.** Rate of oxidation of *iso*butane as suggested in [91].

In another study, Cavani et al. [28] found that the conversion of *iso*butane over $(NH_4)_4PMo_{11}VO_{40}$ increased with increasing the contact time from 1 to 5 s at 320 °C. In this case, due to the presence of V, the selectivity to MAA remained essentially constant or even increased slightly, despite the consequent decrease in MAC selectivity. In other words, while MAC decomposes to AA and $CO_x$, MAA is also formed. This can also be explained by the formal reaction network proposed by Paul et al. [75], as shown in Scheme 3.

The latter claims that MAA can be formed by both direct and indirect ways—in the latter case, through MAC as an intermediate. The corresponding kinetic model indicates that the rate constants $K_3$ and $K_4$ for the conversion of MAC to MAA and decomposition products are similar, being 50 times that of $K_1$ and $K_2$. This means that prolonging the contact time to obtain *iso*butane conversion will come at the expense of the selectivity of the desired products, since the alkane activation remains the rate-determining step.

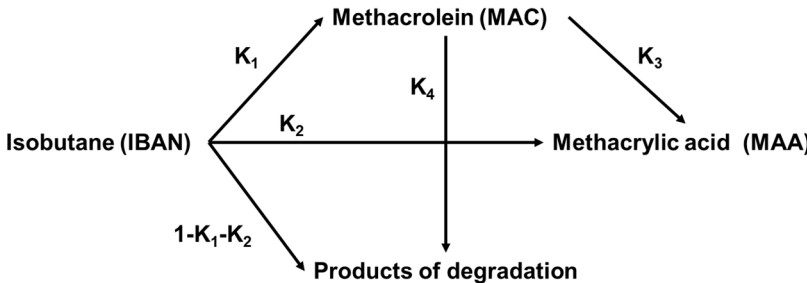

**Scheme 3.** Kinetic reaction schemes for the oxidation of *iso*butane to MAA and MAC as suggested in [75].

## 4. Conclusions and Perspectives

Selective oxidation of *iso*butane for the synthesis of MAA and MAC has induced extensive research interest in the last 40 years. This reaction requires C–H activation, oxidative dehydrogenation, and oxygen insertion. Therefore, the development of the catalytic system must optimize the rate-determining parameters of C–H bond activation in terms of catalyst design and reaction condition optimization. Based on the above discussion, Keggin-type HPCs catalysts have turned out to be the most promising and active ones due to their acidic and redox properties. The acidity and redox properties can be adjusted by substituting pure HPA with cesium, ammonium, vanadium, and other metal ions so as to improve the catalytic performance in terms of *iso*butane conversion and MAA + MAC selectivity.

Insertion of counter-cations, such as $Cs^+$ or/and $NH_4^+$ with $H^+$ in the secondary structure of HPA, improved the catalytic performance, notably due to improved textural properties. As an example, the formation of alkaline salt by inserting cesium atoms leads to the formation of micro- and mesopores.

Introduction of vanadium ($V^{5+}$) in the Keggin unit of phosphomolybdic acid increased the selectivity to MAA by accelerating the oxidation of MAC to MAA, and suppressing the further oxidation of MAC and MAA to $CO_x$. The highest oxidation state of vanadium ($V^{5+}$) increased the redox property of the catalyst by reducing $Mo^{6+}$ to $Mo^{5+}$. Furthermore, vanadium migrates from the primary structure to the secondary one, promoting then the transformation of adsorbed oxygen into lattice oxygen $O^{2-}$.

Supporting of Keggin-type HPCs increased the surface acid site density and stabilized the active phase, hence improving the catalyst performance. However, the catalytic activity and the stability of the active phase are sensitive to the nature of the support. Notably, inert mesoporous materials have many advantages as catalyst supports due to their unique textural properties, such as a high specific surface area, a large pore volume, and a uniform pore size distribution. More strikingly, when CPM is used as a support, the HPCs on the CPM are dispersed in multiple layers, allowing the catalytic performance of the supported catalyst to be close to that of the bulk HPCs themselves, resulting in higher activity.

Figure 11a shows the catalytic performance of all the HPC catalysts cited in this review. The MAC+MAA selectivity and the *iso*butane conversion give a clear view of the state of the art for this research. The conversion of *iso*butane is mainly in the range of 4–12%, while the selectivity to the desired product is more in the 40–60% range. Only some individual catalysts, such as $Cs_2Te_{0.3}(VO)_{0.1}H_xPMo_{12}O_{40}$, show both high *iso*butane conversion (16%) and good selectivity to the desired products (71%).

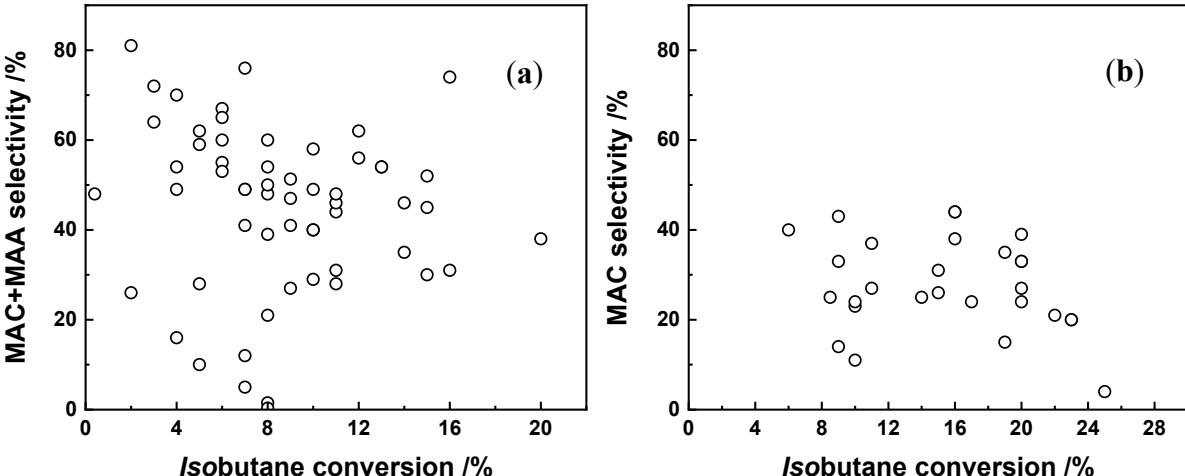

**Figure 11.** MAC+MAA selectivity versus *iso*butane conversion for all HPCs (**a**) and metal oxide (**b**) catalysts in this review.

Additionally, metal oxides are used for this reaction, usually with constituent elements containing V and Mo, and transition metals as dopants. Mixed metal oxides generally have good thermal stability and redox properties. Figure 11b summarizes the catalytic performance of all the metal oxide catalysts cited in this review. Compared to HPCs, mixed metal oxides are able to reach higher temperatures, and exhibit higher *iso*butane conversions (8–24%). However, for the desired products, mainly MAC is yielded, with barely any MAA.

The feed composition, reaction temperature, and contact time are the main factors that affect the catalytic performance. The best catalytic activity was always obtained under *iso*butane-rich conditions. In all cases, the conversion of *iso*butane increases with temperature; however, the selectivity to MAA and MAC typically decreases with temperature due to over-oxidation of the products, since C-H activation remains the rate-determining step.

With respect to the facile over-oxidation of MAA and MAC, further investigation into the selective oxidation of *iso*butane should consider the decoupling of the oxidation and reduction step during reaction, according to the Mars and van Krevelen model. Therefore, several reactor configurations, such as pulse reactors and moving bed reactors, could be envisaged. In a pulsed reactor, the temporal separation of catalyst reduction and reoxidation and the control of the oxygen concentration in the gas phase can be achieved by alternating pulses of reactants and oxygen, whereas the moving bed enables a spatial separation of the two steps. The temporal and spatial separation of redox processes is possible, as can be seen based on reports of the application of this type of reactor to redox processes [92–95], such as the selective oxidation of *n*-butane to maleic anhydride and the oxidation of propane to acrylic acid, etc. Pulse reaction can solve the problem of catalyst cycle regeneration, avoiding over-oxidation of the product. Even better, FCC, such as fluidized-bed reactors, are more efficient at heat transfer in such a way that the reactor may operate *iso*thermally.

**Funding:** This work has benefited from the support of the CSC-Centrale Lille PhD scholarship program.

**Acknowledgments:** Chevreul Institute (FR 2638), Ministère de l'Enseignement Supérieur, de la Recherche et de l'Innovation, Hauts-de-France Region and FEDER are also acknowledged.

**Conflicts of Interest:** The authors declare no conflict of interest.

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
