# Peer review of "Selective Oxidation of Isobutane to Methacrylic Acid and Methacrolein: A Critical Review"

_catalysts, doi:10.3390/catal11070769_

Round 1
Reviewer 1 Report
Authors presented rather interesting review concerning selective catalytic oxidation of isobutane to methacrolein (MAC) and methacrylic acid (MAA). Catalysts based on heteropolycompounds (HPCs) with Keggin structure are mainly considered. These catalysts are compared with alternative metal oxide systems. The information provided is quite complete and useful for researchers in this field. I find the work presented interesting and useful, so it can be published in the Catalysts journal. However, I have a number of questions and comments.
- The authors presented data on the effect of the operating conditions of the catalysts (water additives, temperature, reagent concentration) on their activity and selectivity. According to the presented data, the catalytic properties of HPCs in this process strongly depend on the acidity of the catalyst surface. Therefore, it would be very desirable to provide data on the acid sites of the surface of the catalysts (Brensted or Lewis acid sites, the strength of the sites, etc.). This may help in determining the nature of the catalytically active sites.
- In my opinion, the authors have not fully presented comparative data on metal oxide catalysts of this process. This is an important part of the review, as it is necessary to prove the advantages of HPCs over oxide catalysts.
- The literature data on oxide catalysts should also be presented in the form of tables.
Reviewer 2 Report
Your manuscript entitled „Selective oxidation of isobutane to methacrylic acid and meth- acrolein – a critical review” summarizes the results of studies devoted to the use in the recent years the Keggin-type HPCs to catalyze the oxidation of isobutane to MAA and MAC, and to review alternative metal oxides with proper redox properties for the same reaction. In addition, the influence of the main reaction conditions are also presented. You discuss the results reported by a number of research groups that have implemented various projects for over 25 years. In fact, most of the references cited (90 in total) are papers from the last 15 years. Your manuscript is organized very logically and written clearly. That is why I recommend this manuscript for publication in Catalysts . Although I do not feel entitled to rate English, I am convinced that English and style are good enough and fulfill all editorial requirements
Reviewer 3 Report
In my opinion, this review manuscript addresses a nice view of the subject and is well organized and will be of interest for a wide audience.
I found no issues except one. In Figure 7, I cannot understand how the third apex of the ternary graph, denoted "inert", is a property and how it allows one to read the graph. I know this is adapted from ref. 28 (I have read it) and still do not understand what "inert" means. So, maybe a slight explanation on this would be welcome.
Based on this, minor revision is recommended.
Author Response
please refer to the attachement

Round 2
Reviewer 1 Report
The authors corrected the article according to my comments. I am satisfied with the revised version and recommend the publication of the article.